# PRUNING AS A DEFENSE: REDUCING MEMORIZATION IN LARGE LANGUAGE MODELS

**Mansi Gupta, Nikhar Waghela, Sarthak Gupta & Shourya Goel**
Vision and Language Group,
Indian Institute of Technology, Roorkee
Roorkee, Uttarakhand, India
{m_gupta, n_waghela, sarthak_g}@ma.iitr.ac.in, shourya_g@cs.iitr.ac.in

**Sanjif Shanmugavelu**
Groq Inc
London W6 0ND, UK
sshanmugavelu@groq.com

## ABSTRACT

Large language models have been shown to memorize significant portions of their training data, which they can reproduce when appropriately prompted. This work investigates the impact of simple pruning techniques on this behavior. Our findings reveal that pruning effectively reduces the extent of memorization in LLMs, demonstrating its potential as a foundational approach for mitigating membership inference attacks.

## 1 INTRODUCTION

Large language models are known to memorize portions of their training data, which poses significant privacy and security risks. Although various studies have explored the extent of memorization in LLMs, most of these efforts are qualitative (Carlini et al. (2021)). A recent approach introduces a method to quantify memorization by examining whether models can recall and complete prompts from their training data verbatim (Carlini et al. (2023)). By feeding prefixes of these prompts into a trained model, we can assess its ability to reconstruct the full prompt, thus enabling quantitative analysis of memorization behavior across different models, datasets, and prompt sizes. This framework for quantifying memorization is the basis of our study. Pruning techniques, widely used in machine learning to reduce computational overhead and mitigate overfitting, have been shown to maintain task performance despite reducing the size and complexity of models (Huang et al. (2024),Sun et al. (2024)). Pruning techniques simplify model architectures while preserving task performance (Huang et al. (2024), Sun et al. (2024)). Beyond efficiency, previous studies suggest that pruning can reduce overfitting and induce unlearning by removing memorized information (Pochinkov & Schoots (2024). Motivated by this, we explore pruning to mitigate memorization in large language models.

We systematically evaluate the impact of layer-wise and global pruning strategies on memorization. Building on findings that deeper layers can often be pruned more aggressively without harming performance (Panigrahi et al. (2023), Gromov et al. (2024)), we analyze the effect of pruning specific blocks, including attention layers, to identify those most responsible for memorization. Additionally, we prune at different percentages to assess whether the extent of pruning impacts model performance. Our results show that pruning reduces both computational costs and memorization, providing a lightweight and effective approach to improving privacy in LLMs.

## 2 METHODOLOGY

Drawing reference from this paper (Carlini et al. (2023)), we define a string $s$ as extractable with context $k$ tokens if there exists a $k$ token prefix $p$ such that the concatenation $[p \,||\, s]$ exists in the

training data, and the model $f$ reproduces $s$ when prompted by $p$ using greedy decoding. For example, if a model's training data contains "The capital of Germany is Berlin," and the prefix "The capital of Germany is" results in the output "Berlin," the sequence is deemed extractable with $k = 5$. Our experiments used random subsets of 5000 sequences to estimate intractability at four different context lengths (k = 50,100,200,500) while maintaining statistical confidence efficiently. This is the definition of memorization that we use for all our experiments.

We perform pruning experiments in the Pythia family of models (160M to 12B parameters) (Bengio & LeCun (2007)), trained on the Pile dataset (Gao et al. (2020)). We evaluate memorization across different pruning strategies using sequences sampled from this dataset. We experimented with the following different variants of pruning:

- **Layer-wise Magnitude Pruning:** We individually prune all linear layers within the model, removing the lowest $n\%$ of weights in each layer based on their L1 norm. This localized pruning ensures uniform sparsity across layers while preserving overall model structure.

- **Global Magnitude Pruning:** We prune the lowest $n\%$ of weights across the entire model based on their L1 norm, allowing for more flexible weight removal across layers.
  - *All Linear Layers:* Both MLP and attention layers undergo pruning to reduce redundancy in learned representations.
  - *Attention-Only Layers:* Pruning is applied exclusively to attention layers, keeping MLP layers intact to assess the impact of attention-driven memorization. This allows us to examine the impact of reducing attention capacity while preserving the feedforward network structure.

- **Selective Layer Pruning:** To analyze the role of deeper layers in memorization, we globally prune either the first 25% of layers or the last 25% of layers. This helps determine whether early or later layers contribute more to memorization and overall model performance.

For each pruned variant, we evaluate the degree of memorization across different context lengths and draw reference from some previous work(Chrysostomou et al. (2024),Huang et al. (2024),Liang et al. (2021)) to ensure less performance degradation in pruning LLMs. We experiment with pruning at different levels for varying model sizes, as smaller models are more sensitive to pruning. For each model, we fix two pruning levels, denoted Level 1 and Level 2, which are specific to their size, as shown in the below table.

Table 1: Pruning levels for different Pythia model sizes.

| Model | Level 1 | Level 2 |
|---|---|---|
| Pythia-160M | 10% | 15% |
| Pythia-420M | 15% | 20% |
| Pythia-2.8B | 20% | 30% |
| Pythia-6.9B | 30% | 40% |
| Pythia-12B | 35% | 45% |

We also assess the perplexity of the pruned models to ensure that, despite reduced memorization, they still generate coherent and contextually relevant completions.

## 3 RESULTS

Table 2 reports the percentage of memorized data, calculated as the number of samples correctly reproduced by the model, averaged across 5,000 samples and various context lengths. The results are further averaged over the two pruning levels, with detailed results and perplexity values for each pruned variant presented in Appendix A. The results clearly show that pruning effectively reduces memorization, with higher pruning percentages leading to a more substantial decrease. Notably, global pruning, where all layers are pruned uniformly, reduces memorization more than layer-specific pruning strategies. Among all experiments, pruning only the attention layers achieved the

Table 2: Model-wise Average Fraction of Memorized Data

| Models | Baseline | layer-wise | Global | Attention | First 25% | Last 25% |
|---|---|---|---|---|---|---|
| **Pythia-160m** | 0.0065 | 0.0016 | 0.0012 | 0.0008 | 0.0012 | 0.0010 |
| **Pythia-410m** | 0.01 | 0.0038 | 0.0035 | 0.0021 | 0.0036 | 0.0030 |
| **Pythia-2.8b** | 0.015 | 0.0069 | 0.0048 | 0.003 | 0.0069 | 0.0050 |
| **Pythia-6.9b** | 0.018 | 0.012 | 0.009 | 0.006 | 0.008 | 0.007 |
| **Pythia-12b** | 0.022 | 0.016 | 0.014 | 0.008 | 0.012 | 0.010 |

most significant reduction, indicating that these layers store a considerable amount of information, but they also hurt the performance of the model. Pruning deeper layers also contributes to a notable reduction in memorization, while maintaining performance levels, as substantiated by previous work in this direction (Gromov et al. (2024)).

Table 3 reports the average perplexity values for all models across different context lengths and pruning levels. We evaluate the language model's perplexity across different pruning levels to ensure that pruning does not excessively degrade model performance. Perplexity measures the model's predictive uncertainty, with lower values indicating better language modeling capabilities. By quantifying perplexity, we can establish a threshold that balances the trade-off between reducing memorization and maintaining task performance. This analysis provides insights into how much pruning is acceptable before performance loss outweighs the benefits of reduced memorization, allowing for more informed decisions on pruning strategies and levels.

Table 3: Average perplexity values across both pruning levels

| Models | Baseline | Layer-wise | Global | Attention | First 25% | Last 25% |
|---|---|---|---|---|---|---|
| **Pythia-160m** | 24.41 | 28.45 | 29.16 | 34.84 | 30.21 | 28.99 |
| **Pythia-410m** | 14.78 | 19.63 | 20.17 | 23.39 | 20.41 | 19.93 |
| **Pythia-2.8b** | 9.32 | 9.92 | 10.04 | 12.93 | 11.01 | 9.99 |
| **Pythia-6.9b** | 8.23 | 8.95 | 9.02 | 10.96 | 9.35 | 9.00 |
| **Pythia-12b** | 7.42 | 8.06 | 8.20 | 10.79 | 8.79 | 8.33 |

As shown in Table 3, pruning attention layers results in a higher perplexity, indicating a notable drop in performance. In contrast, pruning deeper layers and applying layer-wise pruning result in a smaller increase in perplexity, suggesting a more balanced trade-off between reducing memorization and preserving model performance.

## 4 LIMITATIONS AND FUTURE WORK

Our study was limited to 5,000 training samples due to computational constraints, leaving scope for testing on larger datasets to gain deeper insights. Additionally, future work could explore other pruning methods, such as in-training pruning (Roy et al. (2020)), to enhance performance stability while reducing memorization. Investigating alternative sparsity techniques, such as quantization and low-rank approximations, could provide further improvements. Expanding the analysis to include different pruning strategies and their effects on various components of model architectures presents a valuable direction for understanding and mitigating memorization in large language models. Also, we have primarily used perplexity as a metric to assess model performance post-pruning. However, future work could incorporate more comprehensive evaluation metrics, such as ROUGE or BLEU scores, to better capture the quality of the generated text.

## 5 CONCLUSION

Our experiments demonstrate that pruning is an effective baseline for mitigating membership inference attacks, as it reduces memorization risk across all context lengths. By introducing sparsity, pruning prevents the exact reproducibility of training data, reduces computational overhead, and maintains model performance, making it a practical solution for addressing memorization in large language models. Pruning attention layers results in the most significant reduction in memorization, although it comes at the cost of a performance drop. This highlights the role of attention layers in

memorization and suggests avenues for future research into targeted mitigation techniques. Additionally, pruning deeper layers results in a substantial reduction in memorization while maintaining performance, making it an efficient baseline strategy. Exploring more adaptive pruning techniques could further illuminate how varying sparsity levels impact memorization.

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

## A    APPENDIX

## PERPLEXITY RESULTS

## RESULTS ACROSS DIFFERENT CONTEXT LENGTHS AND PRUNING LEVELS

We present the results across various context lengths of the input prefix text, reporting the percentage of memorized samples for each model variant, as summarized below.

Table 4: Perplexity values for a lower level of pruning

| Models | Baseline | layer-wise | Global | Attention | First 25% | Last 25% |
|---|---|---|---|---|---|---|
| **Pythia-160m** | 24.41 | 27.45 | 28.18 | 33.45 | 29.19 | 27.90 |
| **Pythia-410m** | 14.78 | 18.43 | 19.65 | 22.14 | 19.90 | 18.78 |
| **Pythia-2.8b** | 9.32 | 9.80 | 9.94 | 12.13 | 10.93 | 9.98 |
| **Pythia-6.9b** | 8.23 | 8.91 | 8.98 | 10.92 | 9.13 | 8.97 |
| **Pythia-12b** | 7.42 | 7.98 | 8.10 | 10.44 | 8.34 | 8.16 |

Table 5: Perplexity values for a higher level of pruning

| Models | Baseline | layer-wise | Global | Attention | First 25% | Last 25% |
|---|---|---|---|---|---|---|
| **Pythia-160m** | 24.41 | 29.45 | 30.14 | 36.23 | 31.23 | 30.07 |
| **Pythia-410m** | 14.78 | 20.82 | 20.68 | 24.63 | 20.92 | 21.08 |
| **Pythia-2.8b** | 9.32 | 10.03 | 10.14 | 13.73 | 11.08 | 9.99 |
| **Pythia-6.9b** | 8.23 | 8.98 | 9.06 | 10.99 | 9.56 | 9.03 |
| **Pythia-12b** | 7.42 | 8.13 | 8.30 | 11.13 | 9.24 | 8.49 |

Table 6: Fraction of Memorization for Pythia-160m

| Context Length | Baseline | Layer Wise | Global | Attention | First 25% | Last 25% |
|---|---|---|---|---|---|---|
| **Lesser Pruning** | | | | | | |
| 50 | 0.008 | 0.0018 | 0.0014 | 0.0009 | 0.0013 | 0.0014 |
| 100 | 0.0075 | 0.0019 | 0.0013 | 0.0010 | 0.0014 | 0.0013 |
| 200 | 0.008 | 0.0020 | 0.0014 | 0.001 | 0.0015 | 0.0011 |
| 500 | 0.0075 | 0.0019 | 0.0012 | 0.0010 | 0.0011 | 0.0012 |
| **Higher Pruning** | | | | | | |
| 50 | 0.005 | 0.0010 | 0.0010 | 0.0005 | 0.0012 | 0.0008 |
| 100 | 0.0055 | 0.011 | 0.0010 | 0.0006 | 0.0011 | 0.0007 |
| 200 | 0.0055 | 0.0014 | 0.0011 | 0.0006 | 0.0012 | 0.0008 |
| 500 | 0.005 | 0.0017 | 0.0012 | 0.0007 | 0.0011 | 0.0008 |

Table 7: Fraction of Memorization for Pythia-410m

| Context Length | Baseline | Layer Wise | Global | Attention | First 25% | Last 25% |
|---|---|---|---|---|---|---|
| **Lesser Pruning** | | | | | | |
| 50 | 0.015 | 0.005 | 0.004 | 0.003 | 0.004 | 0.0036 |
| 100 | 0.016 | 0.004 | 0.0035 | 0.0025 | 0.005 | 0.0035 |
| 200 | 0.014 | 0.005 | 0.005 | 0.003 | 0.004 | 0.0040 |
| 500 | 0.015 | 0.005 | 0.004 | 0.0025 | 0.0035 | 0.0038 |
| **Higher Pruning** | | | | | | |
| 50 | 0.005 | 0.0025 | 0.0035 | 0.0015 | 0.0026 | 0.0023 |
| 100 | 0.006 | 0.0025 | 0.003 | 0.0014 | 0.0026 | 0.0024 |
| 200 | 0.004 | 0.0026 | 0.003 | 0.0016 | 0.0027 | 0.0025 |
| 500 | 0.005 | 0.0028 | 0.0025 | 0.0013 | 0.0027 | 0.0023 |

Table 8: Fraction of Memorization for Pythia-2.8b

| Context Length | Baseline | Layer Wise | Global | Attention | First 25% | Last 25% |
|---|---|---|---|---|---|---|
| **Lesser Pruning** | | | | | | |
| 50 | 0.016 | 0.0075 | 0.005 | 0.003 | 0.0052 | 0.0053 |
| 100 | 0.018 | 0.007 | 0.0052 | 0.0034 | 0.0051 | 0.0052 |
| 200 | 0.015 | 0.0072 | 0.0048 | 0.0036 | 0.0047 | 0.005 |
| 500 | 0.016 | 0.007 | 0.0051 | 0.0032 | 0.0051 | 0.0052 |
| **Higher Pruning** | | | | | | |
| 50 | 0.014 | 0.0065 | 0.0046 | 0.003 | 0.0047 | 0.0047 |
| 100 | 0.015 | 0.0066 | 0.0045 | 0.0024 | 0.0046 | 0.0048 |
| 200 | 0.013 | 0.0068 | 0.0047 | 0.0026 | 0.0048 | 0.0049 |
| 500 | 0.013 | 0.0066 | 0.0045 | 0.0028 | 0.0044 | 0.0049 |

Table 9: Fraction of Memorization for Pythia-6.9b

| Context Length | Baseline | Layer Wise | Global | Attention | First 25% | Last 25% |
|---|---|---|---|---|---|---|
| **Lesser Pruning** | | | | | | |
| 50 | 0.019 | 0.012 | 0.01 | 0.007 | 0.008 | 0.007 |
| 100 | 0.019 | 0.014 | 0.01 | 0.008 | 0.01 | 0.008 |
| 200 | 0.018 | 0.013 | 0.011 | 0.006 | 0.009 | 0.006 |
| 500 | 0.028 | 0.012 | 0.009 | 0.006 | 0.009 | 0.007 |
| **Higher Pruning** | | | | | | |
| 50 | 0.017 | 0.011 | 0.009 | 0.006 | 0.006 | 0.008 |
| 100 | 0.018 | 0.011 | 0.008 | 0.005 | 0.007 | 0.007 |
| 200 | 0.017 | 0.010 | 0.008 | 0.005 | 0.008 | 0.007 |
| 500 | 0.018 | 0.0011 | 0.007 | 0.005 | 0.007 | 0.007 |

Table 10: Fraction of Memorization for Pythia-12b

| Context Length | Baseline | Layer Wise | Global | Attention | First 25% | Last 25% |
|---|---|---|---|---|---|---|
| **Lesser Pruning** | | | | | | |
| 50 | 0.023 | 0.0016 | 0.014 | 0.008 | 0.014 | 0.01 |
| 100 | 0.022 | 0.017 | 0.013 | 0.009 | 0.012 | 0.011 |
| 200 | 0.024 | 0.016 | 0.014 | 0.008 | 0.013 | 0.011 |
| 500 | 0.024 | 0.017 | 0.016 | 0.010 | 0.012 | 0.01 |
| **Higher Pruning** | | | | | | |
| 50 | 0.021 | 0.015 | 0.014 | 0.007 | 0.012 | 0.01 |
| 100 | 0.020 | 0.016 | 0.014 | 0.007 | 0.011 | 0.09 |
| 200 | 0.021 | 0.016 | 0.014 | 0.008 | 0.010 | 0.01 |
| 500 | 0.021 | 0.017 | 0.013 | 0.008 | 0.012 | 0.09 |

