# OpenReview forum: "PRUNING AS A DEFENSE: REDUCING MEMORIZATION IN LARGE LANGUAGE MODELS"
_ICLR.cc/2025/Workshop/BuildingTrust — BuildingTrust_

### Official Review · Reviewer_Fdn8 · 2025-02-20
**This paper investigates the impact of pruning techniques on reducing memorization in large language models (LLMs), addressing a critical issue in privacy and security. The study is well-structured, with clear methodology and comprehensive experiments using the Pythia family of models. The results demonstrate that pruning effectively reduces memorization while maintaining model performance, with global pruning and attention-layer pruning showing the most significant effects. The paper is original, relevant, and contributes valuable insights to the field. However, it is limited by a small dataset and reliance on perplexity as the primary performance metric. Overall, the paper is a strong candidate for acceptance, with potential for further exploration in future work.**

**Rating:** 8
**Confidence:** 4

**Review:**

Quality
The paper is well-structured and presents a thorough investigation into the impact of pruning on memorization in large language models (LLMs). The methodology is sound, and the experiments are well-designed, leveraging the Pythia family of models and the Pile dataset. The results are clearly presented, with tables summarizing the reduction in memorization and the impact on perplexity. The paper also acknowledges its limitations and suggests directions for future work, which adds to its credibility.

Clarity
The paper is generally clear and well-written. The abstract provides a concise overview of the study, and the introduction effectively sets the stage for the research. The methodology section is detailed and explains the pruning strategies and evaluation metrics clearly. However, some parts of the paper could benefit from more explicit explanations, particularly in the results section, where the implications of the findings could be discussed in greater depth.

Originality
The paper addresses a significant and timely issue in the field of LLMs: the memorization of training data and its implications for privacy and security. While pruning is not a new technique, its application to mitigate memorization in LLMs is novel and contributes to the ongoing discourse on model efficiency and privacy. The paper builds on existing work but offers new insights into how pruning can be used to reduce memorization.

Significance
The findings of this paper are highly relevant to the field of machine learning, particularly in the context of privacy-preserving AI. The demonstration that pruning can effectively reduce memorization without significantly degrading model performance is a valuable contribution. This work has the potential to influence future research and practical applications in the development of more secure and efficient LLMs.

Pros
Novel Application: The use of pruning to mitigate memorization in LLMs is innovative and addresses a critical issue in the field.

Comprehensive Experiments: The paper presents a thorough evaluation of different pruning strategies across various model sizes.

Clear Results: The results are well-presented and supported by detailed tables and analysis.

Future Work: The paper acknowledges its limitations and suggests valuable directions for future research.

Cons
Limited Dataset: The study is limited to 5,000 training samples, which may not be representative of larger datasets.

Performance Metrics: The paper primarily uses perplexity to assess model performance. Incorporating additional metrics like ROUGE or BLEU scores could provide a more comprehensive evaluation.

Depth of Discussion: The implications of the findings could be discussed in greater depth, particularly in relation to existing literature and potential real-world applications.

---

### Official Review · Reviewer_HSFE · 2025-03-01
**This review highlights the paper’s experimental design and relevance to AI privacy but notes its limited model scope and absence of comparisons with other privacy techniques.**

**Rating:** 7
**Confidence:** 2

**Review:**

Summary [This paper explores pruning as a method to reduce memorization in large language models (LLMs) and improve privacy. The authors test layer-wise, global, and selective pruning on Pythia models (160M–12B parameters) trained on The Pile dataset. They measure memorization reduction using extractability metrics and analyze the trade-offs between privacy and model performance. The results show that pruning attention layers reduces memorization the most, but also increases perplexity, lowering model accuracy. The study suggests global pruning is more effective than layer-wise pruning and that pruning deeper layers balances memorization reduction and performance.]

Strengths
[-Addresses an important problem: LLM memorization risks privacy violations and security breaches. The paper provides a practical approach to reducing this risk. -Clear experimental design: The study tests multiple pruning methods across various model sizes and evaluates their impact on memorization and performance. -Strong empirical results: The findings show which pruning strategies work best and explain why attention layers store memorized data. -Relevant to AI privacy and safety: This research can help organizations deploying LLMs reduce legal and security risks related to data leakage.]

Weaknesses
[-Limited model and dataset scope: The study only tests Pythia models on The Pile dataset, limiting generalizability to larger models. -Lack of performance vs. privacy trade-off discussion: The study shows that pruning harms accuracy but does not analyze how much trade-off is acceptable in real-world applications. -No comparison with other privacy techniques: The paper does not compare pruning with differential privacy, knowledge distillation, or other defense methods.]

---

### Official Review · Reviewer_JPoU · 2025-03-01
**Great Potential for a future longer submission.**

**Rating:** 7
**Confidence:** 3

**Review:**

## Summary
This paper empirically evaluates the impact of various weight pruning strategies on a language model’s ability to memorize. The authors highlight a tradeoff between model performance and memorization, with certain pruning strategies achieving better balance.

## Strengths
* **Clarity:** The paper clearly defines the problem and describes the experimental setup effectively.
* **Motivation:** The study is well-motivated, addressing a relevant problem with established techniques.
* **Potential:** If the limitations and future directions are addressed, the work has the potential to be a strong 8-page conference submission.

##  Weaknesses
* **Lack of Random baseline:** While some of the pruning strategies seem to be inspired by the literature, the reader could still benefit from understanding why intelligent strategies of pruning is advantageous in memorization. This could be done by introducing a random baseline that prunes random $n\%$ of the weights.
* **Lacking analysis on the tradeoff:** The paper introduces the tradeoff between performance and memorization—high-performing models tend to memorize more. While empirical results hint at this, the evidence for selecting the best tradeoff strategy is insufficient. A stronger case could be made by evaluating each strategy across multiple pruning levels (beyond just two).

## Recommendation
* **Decision**: **Accept**

### Additional Feedback
* Expanding the dataset and incorporating additional evaluation metrics, as noted by the authors, would further strengthen the paper's contributions.

---

### Decision · Program_Chairs · 2025-03-01

Accept